# Impact of Dual-Energy Computed Tomography (DECT) Postprocessing Protocols on Detection of Monosodium Urate (MSU) Deposits in Foot Tendons of Cadavers

**DOI:** 10.3390/diagnostics13132208

**Published:** 2023-06-29

**Authors:** Andrea Sabine Klauser, Sylvia Strobl, Christoph Schwabl, Christian Kremser, Werner Klotz, Violeta Vasilevska Nikodinovska, Hannes Stofferin, Yannick Scharll, Ethan Halpern

**Affiliations:** 1Department for Radiology, Medical University of Innsbruck, 6020 Innsbruck, Austria; andrea.klauser@i-med.ac.at (A.S.K.); christoph.schwabl@tirol-kliniken.at (S.S.); christian.kremser@i-med.ac.at (C.K.); yannick.scharll@i-med.ac.at (Y.S.); 2Department of Internal Medicine, Medical University of Innsbruck, 6020 Innsbruck, Austria; werner.klotz@tirol-kliniken.at; 3University Surgical Clinic “St. Naum Ohridski”, Faculty of Medicine, Ss. Cyril and Methodius University, 1000 Skopje, North Macedonia; v_vasilevska@yahoo.com; 4Institute of Clinical and Functional Anatomy, Medical University of Innsbruck, 6020 Innsbruck, Austria; hannes.stofferin@i-med.ac.at; 5Jefferson Prostate Diagnostic and Kimmel Cancer Center, Department of Radiology and Urology, Thomas Jefferson University, Philadelphia, PA 19107, USA; ethan.halpern@jefferson.edu

**Keywords:** dual-energy computed tomography, gout, tendon, foot, monosodium urate deposits

## Abstract

Objective: To evaluate two different dual-energy computed tomography (DECT) post-processing protocols for the detection of MSU deposits in foot tendons of cadavers with verification by polarizing light microscopy as the gold standard. Material and Methods: A total of 40 embalmed cadavers (15 male; 25 female; median age, 82 years; mean, 80 years; range, 52–99; SD ± 10.9) underwent DECT to assess MSU deposits in foot tendons. Two postprocessing DECT protocols with different Hounsfield unit (HU) thresholds, 150/500 (=established) versus 120/500 (=modified). HU were applied to dual source acquisition with 80 kV for tube A and 140 kV for tube B. Six fresh cadavers (4 male; 2 female; median age, 78; mean, 78.5; range 61–95) were examined by DECT. Tendon dissection of 2/6 fresh cadavers with positive DECT 120 and negative DECT 150 studies were used to verify MSU deposits by polarizing light microscopy. Results: The tibialis anterior tendon was found positive in 57.5%/100% (DECT 150/120), the peroneus tendon in 35%/100%, the achilles tendon in 25%/90%, the flexor halluces longus tendon in 10%/100%, and the tibialis posterior tendon in 12.5%/97.5%. DECT 120 resulted in increased tendon MSU deposit detection, when DECT 150 was negative, with an overall agreement between DECT 150 and DECT 120 of 80% (*p* = 0.013). Polarizing light microscope confirmed MSU deposits detected only by DECT 120 in the tibialis anterior, the achilles, the flexor halluces longus, and the peroneal tendons. Conclusion: The DECT 120 protocol showed a higher sensitivity when compared to DECT 150.

## 1. Introduction

Gout is a common form of crystal-induced arthritis and can lead to the development of tophaceous deposits in the cartilage, bursae, synovial membranes, and tendons [1,2].

Concerning imaging methods [3,4,5,6,7,8,9], DECT is an established method to detect monosodium urate (MSU) deposits in soft tissue adjacent to joints and is increasingly used for the diagnosis and follow-up of gout with a reported sensitivity of 78–100% and specificity of 89–100% [10]. Furthermore, DECT is part of the 2018 gout classification criteria of the American College of Radiology/European League Against Rheumatism (ACR/EULAR) [11,12].

The deposition of MSU in tendons is important because it may lead to reduced strength and increased risk of rupture.

Tendons are soft connective tissues that are composed of closely packed, parallel collagen fiber bundles. Tendons play an essential role in the musculoskeletal system by transferring tensile loads from muscle to bone so as to enable joint motions and stabilize joints. The impact of tendon MSU crystal deposition on musculoskeletal function is currently uncertain. Tendon rupture due to tophus infiltration has been described in patients with chronic gout, although this appears to be an uncommon event and only a small number of studies reported ruptures of the quadriceps, tibialis anterior, and calcaneus tendons [13,14,15,16].

Racide et al. [17] reported that MSU crystal deposits occur within the substance of tendons at the sites of rupture and that urate crystals lead to a reduction in the tensile strength of the tendon [18].

In the last few years, studies concerning extraarticular MSU deposits have become more frequent [19,20]. However, a general consensus regarding a sensitive DECT protocol has not yet been established, as there is ongoing discussion regarding which MSU deposits found by DECT are urate or merely artifacts [21,22,23,24].

Several typical artifacts have been well described in the literature, e.g., finger nails, tendons and vessels [25]. Only a few studies correlated DECT findings with polarizing light microscopy [26]. Given the lack of verification by microscopy of extraarticular MSU deposits in published studies, it remains questionable whether these findings represent true urate or artifact.

The aim of this study was to assess the change of sensitivity with verification by microscopy as gold standard of MSU deposits in foot tendons of cadavers detected with two different DECT postprocessing protocols, comparing DECT 150, an established DECT postprocessing protocol (Hounsfield unit (HU) threshold of 150/500 (min/max)) [27,28] with DECT 120 (modified postprocessing HU threshold of 120/500 (min/max)) at constant kV of Tubes A and B.

## 2. Materials and Methods

### 2.1. Cadavers

A total of 40 embalmed cadavers (15 male; 25 female; median age, 82 years; mean, 80.5; range, 52–99; SD ± 10.9) were examined by DECT to detect the presence and amount of MSU deposits in foot tendons.

Informed consent was provided according to the last wills of the donors, who had donated their bodies to human research studies. All embalmed and fresh cadavers were referred to DECT after death and were in legal custody of the Anatomy institution, Medical University Innsbruck. No medical history was available including gouty arthritis or hyperuricemia.

In addition to the embalmed cadavers, six fresh cadavers (4 male; 2 female; median age, 78; mean, 78.5; range 61–95) were examined by DECT. Two of these cadavers showed the presence of MSU deposits in foot tendons, verified by polarizing light microscopy. The fresh cadavers were referred to DECT within 48 h after death.

The following predefined anatomical sites were assessed for MSU deposits and graded as shown in Table 1: the extensor halluces longus tendon (EHLT), extensor digitorum longum tendon (EDLT), tibialis anterior tendon (TAT), flexor halluces longus tendon (FHL), tibialis posterior tendon (TP), peroneus tendons (PT), achilles tendon (AT), and plantar flexors (PF). To exclude artifacts, MSU crystal characterization using polarized light microscopy was performed for the tibialis anterior, achilles, flexor halluces longus, plantar flexor, and peroneal tendons with positive DECT 120 findings but negative results in the DECT 150 in fresh cadavers.

According to ACR/EULAR guidelines nail bed deposits, submillimeter deposits, skin deposits, and deposits obscured by motion, beam hardening, and vascular artefact were not classified as positive findings in our study [8,12].

### 2.2. DECT

All cadavers were scanned with a 128-row dual-source CT scanner (Somatom Definition Flash; Siemens Healthineers, Forchheim, Germany). Data were acquired at 80/140 kV. For post-processing, the standard protocol using a min/max houndsfield unit (HU) threshold of 150/500 (DECT 150) was compared to a modified protocol with a lowered HU threshold of 120/500 (DECT 120), see Table 2.

The fundamental principle behind the use of DECT is to differentiate materials based on their relative absorption of X-rays at different photon energy levels (typically at 80 and 140 kVp).

A dual-source DECT scanner can perform simultaneous acquisitions at two energy levels (80 and 140 kVp) using two separate X-ray tubes and detectors positioned 90 to 95 degrees apart [20]. The standard protocol settings are 80 kV/100–140 mAs for tube A and 140 kV/200–250 mAs for tube B, with a ratio of 1.36, range 4, minimum HU of 150, and maximum HU of 500. Using independent tube current modulation in combination with iterative reconstruction and integrated circuits within the detector module, high-resolution images with excellent material separation are obtained without an increase in radiation dose [28].

The acquired datasets are reconstructed in the required planes and processed with dual-energy software utilizing a standardized two-material decomposition algorithm designed for specific clinical applications. The gout algorithm is performed to separate MSU from calcium using soft tissue as the baseline. The two-material decomposition algorithm is based on the principle that materials with a high atomic number such as calcium would demonstrate a higher increase in attenuation at higher photon energies than does a material composed of low atomic number materials such as MSU. This difference in attenuation is independent of density or concentration of the material or tissue. Once separated and characterized, the materials are color-coded and overlaid on multi-planar reformatted cross-sectional images.

We chose green pixels for MSU deposits using the software of the Syngovia workstation (Siemens Healthineers, Erlangen, Germany). The post-processing software enables real-time manipulation of the images at source resolution, in any plane and in two- as well as three-dimensions, to best depict the MSU deposits. Preprocessed and processed images are transferred to the picture archiving system (PACS). Corresponding pre-processed grey-scale images are reviewed for presence of deposits.

The software can not differentiate confounding artifacts that should not be included in the detection/quantification of MSU deposits, but an experienced and trained radiologist can differentiate MSU deposits from artifacts, as previously described in the literature.

Scan parameters were 2 × 64 × 0.6 mm at a rotation time of 280 ms, DLP 219 mGycm, CTDI vol 11.5 L, and total mAs 3415. Transverse sections, coronal and sagittal reformations were reconstructed from the DE datasets with an increment of 0.5 mm and a resolution slice thickness of 0.4 mm in soft tissue kernel (D30) and bone kernel (B60). D30 kernel was used for DE processing and MSU detection.

A commercially available picture archiving and communication system (PACS) was used to view the images (IMPAX; Agfa-Gevaert, Mortsel, Belgium).

Two radiologists with 4 and 15 years of experience in gout imaging by DECT evaluated DECT images in consensus. Each cadaver was read as a single observation, either positive or negative. Multiple positive sites in a single tendon were counted as a single observation with the largest size deposit recorded.

### 2.3. Polarizing Microscopic Evaluation

DECT-positive MSU foot tendons from fresh cadavers were dissected for gross anatomic sections obtained at defined anatomic landmarks. Specimens were cut unfixed into 5 × 5 mm sections, embedded using Tissue-Tek^®^ O.C.T.™ (Sakura Finetek USA, Inc., Torrance, CA, USA) compound medium and sectioned at 5 μm using a Leica CM1950 S cryostat (Leica Biosystems, Wetzlar, Germany). After mounting on microscope slides and covering using glycerine/PBS solution, cryostat sections examination was performed with compensated polarized light microscopy (PLM) at 400× magnification. First-order red compensation (X) was performed to identify MSU crystals by their needle-like appearance and strong negative birefringence. The performing pathologist was blinded to radiological results.

### 2.4. Statistical Analysis

Statistical analysis was performed using R Project for Statistical Computing 3.4.1 [29]. The presence of a positive or negative final diagnosis for gout, presence of MSU deposits in foot tendons and grading of tophus size were tabulated for DECT 120 and DECT 150. To analyze the agreement between DECT 120 and 150, contingency tables were created, and the overall agreement was calculated. The agreement of overall positive and negative findings between DECT 120 and 150 was determined using a McNemar’s Chi- square test and quadratic weighted kappa. The relationship between the gradings of MSU deposits obtained with DECT120 and DECT150 was analyzed using a Pearson Chi-square for the symmetry test. In general, results were considered significant for *p*-values less than 0.05.

## 3. Results

In eight cadavers a tendon MSU deposit was identified by DECT 120 detection, while DECT 150 was negative. In the remaining 32 specimens both DECT 120 and DECT 150 agreed in their positive final reading.

Although this resulted in an overall agreement between DECT 150 and DECT 120 of 80%, McNemar´s Chi-square test and quadratic weighted kappa revealed a significant difference in MSU detection between both methods (*p* = 0.013). Table 3 gives the overall agreement between DECT 120 and DECT 150 regarding the MSU grading for all individual tendons and the corresponding *p*-values for the Pearson Chi^2^ test of independence.

The overall agreement between the two methods was relatively low (0–30%), and the difference was statistically significant, except for the achilles tendon. 

The tibialis anterior tendon (TAT), peroneus tendons (PT), and flexor halluces longus tendon (FHL) were the most commonly involved tendons, followed by the tibialis posterior tendon (TP), achilles tendon (AT), and extensor halluces longus tendon (EHLT), as shown in Table 4.

The most common Score 3 MSU deposition site detected by DECT 120 was the achilles tendon followed by the tibialis anterior tendon, Table 5, Figure 1.

A major reason for the low agreement was that MSU deposit size was larger detected by DECT 120 than by DECT 150 (*p* < 0.001) for all tendons leading to different MSU gradings, see Table 5.

In 2/6 fresh cadavers MSU deposits in DECT 120, not visible in DECT 150 could be verified by microscopy in the tibialis anterior, achilles, flexor halluces longus, and peroneal tendons, see Figure 2 and Figure 3.

No false positive MSU interpretations were found on pathologic review of our DECT 120 readings in the fresh cadavers.

## 4. Discussion

Tendon involvement in patients with gout has been evaluated by US and DECT [30,31,32] and it has been shown that tendons are a frequent location of extraarticular MSU deposition in the lower limbs, especially in longstanding gout. DECT demonstrated high sensitivity, specificity, and reproducibility in the detection of MSU deposits in tendons in previous studies [26,31,33,34]. Yuan et al. [3] reported tendons as the most frequent anatomical location site of MSU deposits in a DECT study of 184 joints, regardless of whether the intraarticular DECT results were positive or negative.

In a DECT study of 92 patients with longstanding tophaceous gout, Dalbeth et al. [31] reported that tendons were the most frequent site of MSU deposits showing a high prevalence in the achilles tendon of 39.1%, followed by peroneal tendons in 18.1%. Involvement by MSU was found in 7% and 5%, respectively for the TAT and FHL, which is in contrast to our results where the TAT was involved in 57.5%/100% (DECT 150/120) and AT in 25%/90% (DECT 150/120).

Recently, Dubief et al. [23] described the prevalence of MSU deposits in foot tendons by using different DECT postprocessing protocols. Using the minimum HU 120 setting, they classified most of the findings as artifacts, but they did not have verification by histology as the gold standard, which we performed in our study. DECT 120 showed a better correlation regarding tophus size when compared to the US in previous studies [35].

Dalbeth et al. [36] reported that DECT is a highly reproducible method for measuring urate deposits within tophi, revealing the composition of tophi that contains variable urate deposits embedded within soft tissue where MSU deposits are scattered across the tophus surrounded by soft tissue. These findings are consistent with a previous histological analysis of tophi from the same team.

As reported by Melzer et al. [37], DECT can identify ‘dense’ tophi (with at least 15–20% urate volume in the tophus), but tophi with lower urate volumes may not be detected. This limitation might be overcome by the use of DECT 120, as shown in our study. When only considering DECT 150, we would have missed 20% of the MSU deposits in foot tendons. Lowering the HU threshold without changing scan parameters (tube A and B at 80/140 kV) was helpful to better identify tophi with low MSU concentration with verification by histology.

Though DECT 120 shows great potential, it is not without limitations. It has to be noted that DECT 120 is more susceptible to artifacts and an increased amplification of noise, see Figure 4.

Therefore, imaging results should be interpreted with caution by an experienced radiologist who is well-trained to recognize artifacts. Locations notorious for artifacts are nose, skin, calluses, and nail beds [4,24,25] which were obviously more prevalent and increased in size when using DECT 120 (Figure 1).

Furthermore, MSU deposits shown in DECT 120 and not visible in DECT 150 were only histologically verified in fresh cadavers not in embalmed cadavers because of potential interference with the fixation process in embalmed cadavers. This is a clear limitation of our study, as we cannot clearly prove that some of the positive observations at DECT 120 were not false positives. Nonetheless, our experience with the six fresh cadavers suggests that the interpretation of our experienced observers did not yield a significant number of false positive results, as all of their positive interpretations in the fresh specimens were confirmed by polarized light microscopy.

An additional limitation of this study is the lack of assessment of inter-observer agreement, and the lack of a medical history for the cadavers. Finally, note that our ex-vivo results need to be repeated in live patients to demonstrate clinical applicability. 

We are fully aware that the results we report do not agree with many existing publications on tendon involvement in gout. However, there are individual publications that show similarly high numbers of MSU deposits in tendons. It should be noted that most studies worked with higher cut-off values regarding the HU, and many regions of the tendons were not classified as true MSU but as artifacts. Our study is the first to examine these regions which were previously presumed to represent artifacts utilizing histological processing. The rate of 80% MSU deposits in tendons is strikingly high and raises many questions regarding the composition and nature of tendons, as well as the pathomechanism and clinical relevance of tendon MSU deposits in gout.

In addition to assisting with the diagnosis of gout, DECT may help to monitor the efficacy of urate-lowering therapy and to prove therapeutic outcomes not only in joints but also in tendons [38].

## 5. Conclusions

The DECT 120 protocol showed a higher sensitivity when compared to DECT 150.

## Figures and Tables

**Figure 1 diagnostics-13-02208-f001:**
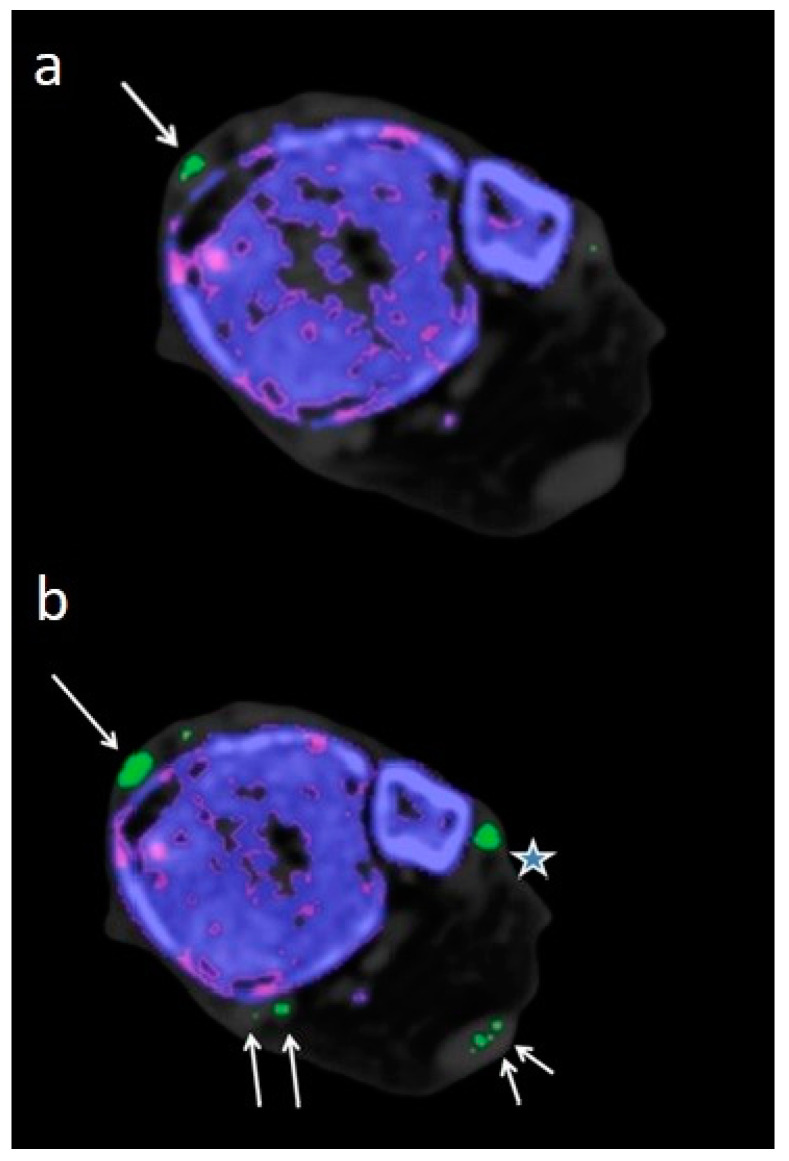
85 year-old-male embalmed cadaver. (**a**) Axial DECT 150 image with Score 1 MSU deposits in tibialis anterior tendon (large arrow). (**b**) Corresponding axial DECT 120 image showing an increased size (Score 2) of the MSU deposit in tibialis anterior tendon (large arrow). MSU deposits were also detected in peroneal tendons (star), achilles tendon (small arrows), and tibialis posterior/flexor halluces longus tendon (middle sized arrows).

**Figure 2 diagnostics-13-02208-f002:**
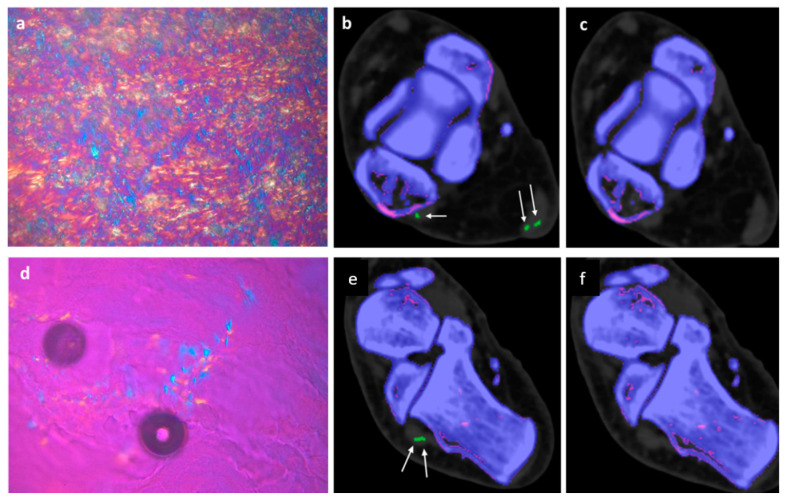
77 year-old-male fresh cadaver. (**a**) Polarizing light microscopic evaluation of a sample taken from achilles tendon, showing large diffuse packed and patchy MSU crystals with strong negative birefringence (bluish structures). (**b**) Corresponding DECT 120 image showing large MSU deposits in the achilles tendon (large white arrows) and one small MSU deposit (small white arrow) in peroneus tendon. (**c**) Corresponding DECT 150 image without evidence of MSU deposits in the achilles and peroneus tendon. (**d**) Polarizing light microscopic evaluation of a sample taken from peroneus tendon, showing diffuse packed and patchy MSU crystals with strong negative birefringence (bluish structures). (**e**) Corresponding DECT120 image showing MSU deposits in peroneal tendons (white arrows). (**f**) Corresponding DECT 150 image without any MSU deposits in peroneal tendons.

**Figure 3 diagnostics-13-02208-f003:**
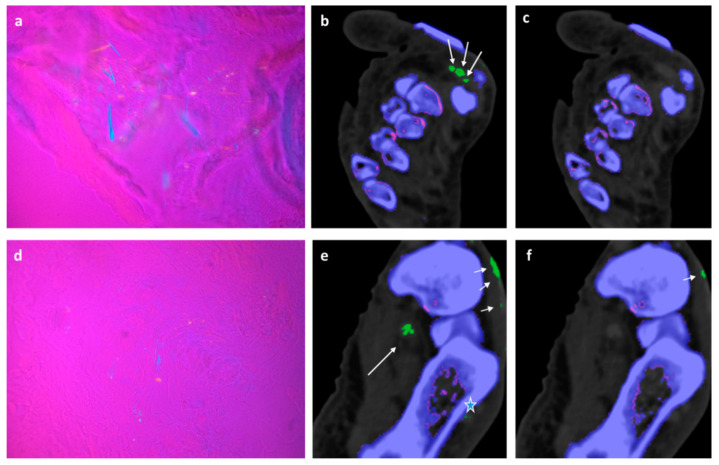
77-year-old-male fresh cadaver. (**a**) Polarizing light microscopic evaluation of a sample taken from flexor halluces longus tendon, showing diffuse packed and patchy MSU crystals with strong negative birefringence. (**b**) Corresponding DECT 120 image showing large MSU deposits in the flexor halluces longus tendon of Digitus 1 (large white arrows). (**c**) Corresponding DECT 150 image without showing any MSU deposits in the flexor halluces longus tendon (**d**) Polarizing light microscopic evaluation of a sample taken from the plantar flexor tendon, showing diffuse packed and patch MSU crystals with strong negative birefringence. (**e**) Corresponding DECT 120 image showing large MSU deposits in plantar flexor tendon with skin artifacts (small white arrows), MSU deposits in the flexor halluces longus tendon (large white arrows) and small MSU deposit in peroneus tendon (STAR). (**f**) Corresponding DECT 150 image without showing any MSU deposits in plantar flexor tendon with skin artefact (small arrow).

**Figure 4 diagnostics-13-02208-f004:**
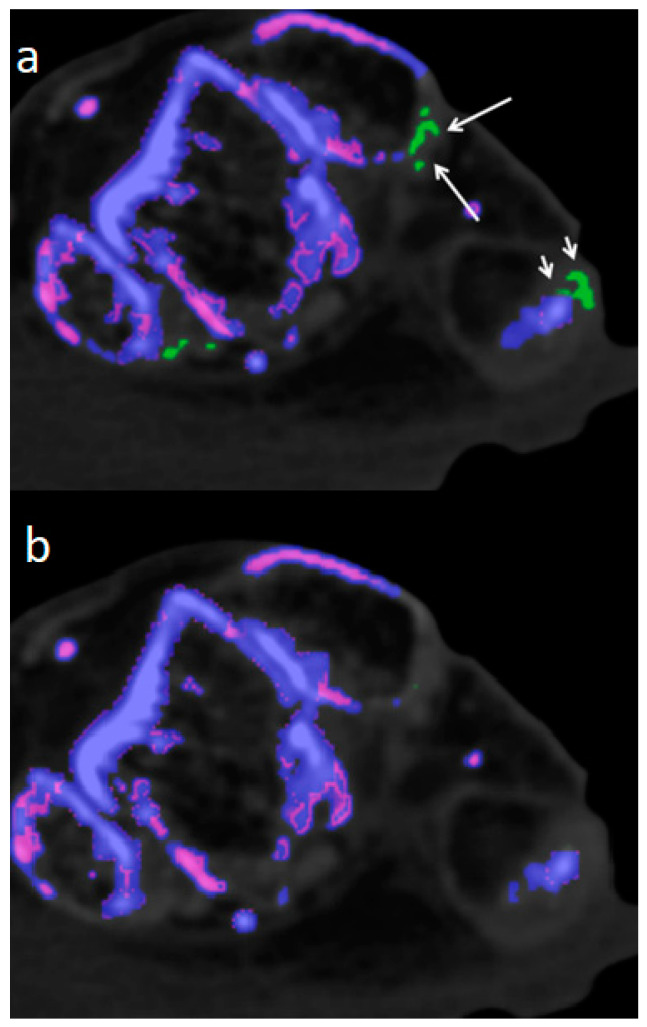
70 year-old-male embalmed cadaver. (**a**) Artifacts shown at distal fibula (long arrows) and calcaneus (small arrows) in DECT 120. (**b**) Corresponding axial DECT 150 image without any artifacts.

**Table 1 diagnostics-13-02208-t001:** Grading of MSU deposits.

Score	MSU Deposit Size
0	MSU absent
1	2–5 mm
2	5–10 mm
3	≥10 mm

**Table 2 diagnostics-13-02208-t002:** Workstation settings of DECT.

DECT Workstation Postprocessing Settings	Tube A		Tube B
	DECT 150	DECT 120	
Soft tissue (HU)	50	50	50
Ratio	1.36	1.36	1.36
Range	4	4	4
Threshold (HU)	150	120	500
Air distance/bone distance	5	5	10

**Table 3 diagnostics-13-02208-t003:** Overall agreement of DECT 150 and DECT 120 in different tendons of embalmed cadavers.

Anatomical Location	Overall Agreement (%)	Quadratic Weighted Kappa	Chi^2^-Test
EHLT	27.5	0.26	*p* = 0.0015
EDLT	30	0.26	*p* = 0.016
TAT	5	0.31	*p* = 0.0005
FHL	0	0.17	*p* = 0.001
TP	5	0.17	*p* = 0.0255
PT	5	0.21	*p* = 0.0005
AT	20	0.19	*p* = 0.13
PF	15		*p* = 0.001

Note: EHLT—extensor halluces longus tendon, EDLT—extensor digitorum longus tendon, TAT—tibialis anterior tendon, FHL—flexor halluces longus tendon, TP—tibialis posterior tendon, PT—peroneus tendons, AT—achilles tendon, and PF—Plantar flexor tendon.

**Table 4 diagnostics-13-02208-t004:** Number of embalmed cadavers when MSU deposits were detected for DECT 150 (first row) and DECT 120 (second row).

Anatomical Location	DECT	Positive
EHLT	150	20/40 (50%)
	120	36/40 (90%)
EDLT	150	10/40 (25%)
	120	35/40 (75%)
TAT	150	23/40 (57.5%)
	120	40/40 (100%)
FHL	150	4/40 (10%)
	120	40/40 (100%)
TP	150	5/40 (12.5%)
	120	39/40 (97.5%)
PT	150	14/40 (35%)
	120	40/40 (100%)
AT	150	10/40 (25%)
	120	36/40 (90%)
PF	150	3/40 (7.5%)
	120	35/40 (87.5%)

Note: EHLT—extensor halluces longus tendon, EDLT—extensor digitorum longus tendon, TAT—tibialis anterior tendon, FHL—flexor halluces longus tendon, TP—tibialis posterior tendon, PT—peroneus tendons, AT—achilles tendon, and PF—Plantar flexor tendon.

**Table 5 diagnostics-13-02208-t005:** Comparison of DECT 150 (first row) and DECT 120 (second row) in foot tendons of embalmed cadavers with regard to Score 0–3.

Anatomical Location	DECT	Score 0	Score 1	Score 2	Score 3
EHLT	150	20/40 (50%)	19/40 (47.5%)	1/40 (2.5%)	0/40 (0%)
	120	4/40 (10%)	19/40 (47.5%)	14/40 (35%)	3/40 (7.5%)
EDLT	150	30/40 (75%)	9/40 (22.5%)	1/40 (2.5%)	0/40 (0%)
	120	5/40 (12.5%)	30/40 (75%)	3/40 (7.5%)	2/40 (5%)
TAT	150	17/40 (42.5%)	17/40(42.5%)	6/40 (15%)	0/40 (0%)
	120	0/40 (0%)	11/40(27.5%)	23/40 (57.5%)	6/40 (15%)
FHL	150	36/40 (90%)	3/40 (7.5%)	1/40 (2.5%)	0/40 (0%)
	120	0/40 (0%)	29/40 (72.5%)	9/40 (22.5%)	2/40 (5%)
TP	150	35/40 (87.5%)	4/40 (10%)	1/40 (2.5%)	0/40 (0%)
	120	1/40 (2.5%)	28/40 (70%)	9/40 (22.5%)	2/40 (5%)
PT	150	26/40(65%)	13/40 (32.5%)	1/40 (2.5%)	0/40 (0%)
	120	0/40 (0%)	21/40 (52.5%)	17/40 (42.5%)	2/40 (2.5%)
AT	150	30/40 (75%)	9/40 (22.5%)	1/40 (2.5%)	0/40 (0%)
	120	4/40 (10%)	21/40 (52.5%)	8/40 (20%)	7/40 (17.5%)
PF	150	37/40 (92.5%)	1/40 (2.5%)	2/40 (5%)	0/40 (0%)
	120	5/40 (12.5%)	30/40 (75%)	3/40 (7.5%)	2/40 (2.5%)

Note: EHLT—extensor halluces longus tendon, EDLT—extensor digitorum longus tendon, TAT—tibialis anterior tendon, FHL—flexor halluces longus tendon, TP—tibialis posterior tendon, PT—peroneus tendons, AT—achilles tendon, PF—Plantar flexor tendon; Score 0 = absent MSU deposits, Score 1 = MSU deposits 2–5 mm, Score 2 = MSU deposits 5–10 mm, and Score 3 = MSU deposits > 10 mm.

## Data Availability

Data supporting reported results may be provided upon reasonable request.

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
