# Peer review of "Impact of Dual-Energy Computed Tomography (DECT) Postprocessing Protocols on Detection of Monosodium Urate (MSU) Deposits in Foot Tendons of Cadavers"

_diagnostics, 2023, doi:10.3390/diagnostics13132208_

Round 1
Reviewer 1 Report
Although there have been many similar studies conducted, what distinguishes this research is its utilization of cadavers to extract components for analysis. In contrast to other human trials where obtaining specimens for confirmation is not possible, this study achieves a higher level of accuracy. The study exhibits a high level of completeness and a well-structured framework, making it an excellent research endeavor.
Author Response
Dear reviewer, thank you very much for your kind words, I am very glad that the work is positively received.
Kind regards!
-----------------------------------------------------------------
Reviewer 2 Report
Dear authors,
In this paper, you sought to compare two different DECT protocols for detection of MSU deposits in cadaveric specimens. Please find my comments below.
Title: The word ‘in’ should be replaced with ‘on’.
Abstract
Line 14: Please spell out DECT.
Line 21: Do 2/6 fresh cadavers represent a sufficient sample size? Please clarify the way you calculated sample size in the methods section.
Introduction
Please re-organise this section appropriately as it appears that there are many paragraphs.
Line 35: Please spell out MSU.
Line 42: Tendon properties should be mentioned more professionally here. For example, you may state that they transmit tensile loads from muscle to bone, they also enable the muscle belly to be at an optimal distance from the joint without an extended length of muscle between origin and insertion, and the fact that they store of energy.
Line 67: Please state where the experiment was conducted (eg which University lab).
For Table 2, you will need to spell out the abbreviations that you used.
Methods
Line 134: Why only two radiologists? Please justify.
Discussion
Line 301: Please avoid using terms like ‘astonishing’ in the text as the readers should be the ones who would judge the result of this work.
Minor language editing is needed.
Author Response
Title: The word ‘in’ should be replaced with ‘on’.
Dear Reviewer, thank you very much, we replaced “in” with “on”.
Abstract
Line 14: Please spell out DECT.
We spelled out DECT.
Line 21: Do 2/6 fresh cadavers represent a sufficient sample size? Please clarify the way you calculated sample size in the methods section.
Of course, the number of fresh cadavers was rather small, due to availability, but for these we took several samples from different tendons, as described in the methods, so that we had a sufficient number of tendon samples, although these were also not from many different cadavers.
Furthermore, the purpose of the fresh cadaver was only to evaluate the tophi positive in the 120 and negative in the 150. The general evaluation of the distribution of MSU deposits was performed in the embalmed cadavers.
Introduction
Please re-organise this section appropriately as it appears that there are many paragraphs.
Dear Reviewer, you are right – we reorganized the paragraph.
Line 35: Please spell out MSU.
We spelled out “MSU”.
Line 42: Tendon properties should be mentioned more professionally here. For example, you may state that they transmit tensile loads from muscle to bone, they also enable the muscle belly to be at an optimal distance from the joint without an extended length of muscle between origin and insertion, and the fact that they store of energy.
Thank you for your advice, we added a paragraph!
Line 67: Please state where the experiment was conducted (eg which University lab).
We added the University!
For Table 2, you will need to spell out the abbreviations that you used.
We spelled out “HU” previously to the Table.
Methods
Line 134: Why only two radiologists? Please justify.
In our institute, only 2 radiologists currently deal in depth with DECT in gout imaging, so the evaluation was performed by these two because of their expertise in this field.
Discussion
Line 301: Please avoid using terms like ‘astonishing’ in the text as the readers should be the ones who would judge the result of this work.
You are totally right, thank you, we deleted those phrases.